

# Evidence of episodic positive selection in *Corynebacterium diphtheriae* complex of species and its implementations in identification of drug and vaccine targets

Marcus Vinicius Canário Viana[1,2], Rodrigo Profeta[1], Janaína Canário Cerqueira[1], Alice Rebecca Wattam[3], Debmalya Barh[1,4], Artur Silva[2] and Vasco Azevedo[1]

[1] Departamento de Genética, Ecologia e Evolução, Universidade Federal de Minas Gerais, Belo Horizonte, Minas Gerais, Brazil
[2] Departamento de Genética, Universidade Federal do Pará, Belém, Pará, Brazil
[3] Biocomplexity Institute, University of Virginia, Charlottesville, Virginia, United States
[4] Institute of Integrative Omics and Applied Biotechnology, Nonakuri, West Bengal, India

Corresponding author
Vasco Azevedo, vasco@icb.ufmg.br

## ABSTRACT

**Background:** Within the pathogenic bacterial species *Corynebacterium* genus, six species that can produce diphtheria toxin (*C. belfantii*, *C. diphtheriae*, *C. pseudotuberculosis*, *C. rouxii*, *C. silvaticum* and *C. ulcerans*) form a clade referred to as the *C. diphtheria* complex. These species have been found in humans and other animals, causing diphtheria or other diseases. Here we show the results of a genome scale analysis to identify positive selection in protein-coding genes that may have resulted in the adaptations of these species to their ecological niches and suggest drug and vaccine targets.

**Methods:** Forty genomes were sampled to represent species, subspecies or biovars of *Corynebacterium*. Ten phylogenetic groups were tested for positive selection using the PosiGene pipeline, including species and biovars from the *C. diphtheria* complex. The detected genes were tested for recombination and had their sequences alignments and homology manually examined. The final genes were investigated for their function and a probable role as vaccine or drug targets.

**Results:** Nineteen genes were detected in the species *C. diphtheriae* (two), *C. pseudotuberculosis* (10), *C. rouxii* (one), and *C. ulcerans* (six). Those were found to be involved in defense, translation, energy production, and transport and in the metabolism of carbohydrates, amino acids, nucleotides, and coenzymes. Fourteen were identified as essential genes, and six as virulence factors. Thirteen from the 19 genes were identified as potential drug targets and four as potential vaccine candidates. These genes could be important in the prevention and treatment of the diseases caused by these bacteria.

## INTRODUCTION

The genus *Corynebacterium* are gram-positive bacteria of biotechnological, medical, and veterinary importance (*Bernard & Funke, 2015*). Within the pathogenic species, some can produce diphtheria toxin (DT) after lysogenization by *tox*+ corynephages (*Bernard & Funke, 2015*). Three species that compose a clade were initially described as potential diphtheria toxin (DT) producers: *C. diphtheriae*, *C. ulcerans* and *C. pseudotuberculosis* (*Bernard & Funke, 2015*). The number of species in the clade of potential DT producers increased to six with the inclusion of the recently described *C. belfantii* (*Dazas et al., 2018*), *C. rouxii* (*Badell et al., 2020*) and *C. silvaticum* (*Dangel et al., 2020*). Those six species are described here as the "*C. diphtheria* complex" (*Badell et al., 2020*).

*C. diphtheriae*, *C. belfantii* and *C. rouxii* infect mainly humans (*Bernard & Funke, 2015*; *Dazas et al., 2018*; *Badell et al., 2020*). *C. ulcerans*, *C. pseudotuberculosis* and *C. silvaticum* infect mainly wild and domesticated mammals and/or can cause zoonosis (*Bernard & Funke, 2015*; *Dangel et al., 2020*). *C. belfantii* and *C. rouxii* have recently been reclassified species from some of the *C. diphtheriae* biovar Belfanti strains (*Dazas et al., 2018*; *Badell et al., 2020*). *C. ulcerans*, *C. pseudotuberculosis* and *C. silvaticum* infect mainly wild and domesticated mammals but can also be zoonotic (*Bernard & Funke, 2015*; *Dangel et al., 2020*).

The *C. diphtheria* complex have an impact on public health, and also on the production of animal-based foods. Some of the species contain both DT and strains that lack the toxin. DT-producing *C. diphtheriae* strains cause cutaneous and respiratory diphtheria (*Zasada, 2013*; *Grosse-Kock et al., 2017*). The report of multidrug-resistant strains from Brazil is a new concern (*Zasada, 2014*; *Hennart et al., 2020*). DT-producing *C. pseudotuberculosis* from biovar equi causes Oedematous Skin Disease in buffalos (*Selim et al., 2015*). *C. ulcerans* infects a broad range of mammal species and DT-producing strains have caused diphtheria (*Hacker et al., 2016*). Some non-DT producing strains of *C. diphtheriae* cause endocarditis, septic arthritis, osteomyelitis and sepsis in humans (*Zasada, 2013*; *Grosse-Kock et al., 2017*). Non-DT producing strains of *C. ulcerans* are associated with ulcers in humans (*Hacker et al., 2016*). *C. pseudotuberculosis* also contains non-DT strains, with those in biovar equi causing ulcerative lymphangitis in horses, and those in the biovar ovis causing caseous lymphadenitis in goat and sheep, and lymphadenitis and abscesses in humans (*Selim et al., 2015*).

There are also *C. diphtheria* complex species that never produce DT but do cause disease. *C. belfantii* causes laryngitis and bronchopathy (*Dazas et al., 2018*). *C. rouxii* causes chronic arteritis leading to ulcerations on feet and legs, and peritonitis (*Badell et al., 2020*). *C. silvaticum* has only been isolated from pigs and roe deer to date, causing caseous lymphadenitis (*Dangel et al., 2020*), and is cytotoxic for human epithelial cells (*Möller et al., 2021*).

The host ranges and virulence mechanisms of these species are not entirely known, and better understanding of their biology could be helpful in controlling this group of pathogens. Diphtheria outbreaks were reported globally between 1921 and 2018. The disease is still endemic in some countries, with thousands of annual cases reported in

Asia and Africa. The disease can emerge when the recommended vaccination programs are not applied or sustained (*Sharma et al., 2019*). The current vaccine is based on the DT toxoid (*Rappuoli & Malito, 2014*) but does not prevent the colonization, transmission, and disease manifestation. In addition, the acquired immunity has been found to decrease with time (*Truelove et al., 2020*). Isolation of symptomatic individuals, antitoxin and antibiotics are still essential in the control of these diseases (*Truelove et al., 2020*). Furthermore, the diversity of DT toxin sequences across strains could reduce the effectiveness of diphtheria toxoid–based vaccines and diphtheria antitoxins (*Otsuji et al., 2019*). Another factor to consider in the control of these pathogens is that non-DT producing strains can cause other diseases, such as ulcers and caseous lymphadenitis, the latter associated with the Phospholipase D toxin produced by *C. ulcerans*, *C. pseudotuberculosis* and *C. silvaticum* (*Bernard & Funke, 2015*; *Dangel et al., 2020*).

Adaptive mutations for a specific ecological niche can be identified using genomic analyses, including genome-scale positive selection analysis (*Kopac et al., 2014*). At an ecological level, routine selection favors the maintenance of a stable population structure over time, while episodic selection is the effect of a sudden environmental disturbance, such as host change (*Brasier, 1995*). At the molecular level, positive selection can help fix adaptive mutations (*Anisimova & Liberles, 2012*). Episodes of positive selection can act on specific codons at specific times (phylogenetic branches), for which branch-site statistical models were developed (*Zhang, 2005*). Information on the amino acids under selection could be used for drug design (*Farhat et al., 2013*), or even reverse vaccinology if the amino acids are surface exposed (*Goodswen, Kennedy & Ellis, 2018*).

In this work, we used a genome scale positive selection analysis to identify the genes that could be involved in ecological adaptation and identified genes that can be used to develop preventive or therapeutic strategies against this group of important pathogens.

## MATERIALS AND METHODS

### Samples and taxonomy

Episodic positive selection across species or other phylogenetic groups of the diphtheriae group was investigated using a branch-site test (*Zhang, 2005*). This test is more appropriate for inter-specific samples, because it assumes that the observed mutations have already been fixed by selection (*Kryazhimskiy & Plotkin, 2008*; *Anisimova & Liberles, 2012*; *Kosiol & Anisimova, 2012*), and one genome could represent a species. For this reason, we limited the samples to one per species, subspecies or biovar. For the foregrounds (target groups), we selected the six type strains from the *C. diphtheria* complex and other strains to represent biovars and lineages. *C. diphtheriae* biovars could not be used as foregrounds as they are united in a single clade (*Sangal et al., 2014*). As background, we selected 40 total genomes that included 30 representative (*O'Leary et al., 2016*), eight complete and three WGS genomes of *Corynebacterium* species, all of which were available in the Pathosystems Resource Integration Center (PATRIC) (*Davis et al., 2020*). These were annotated by RASTtk (*Brettin et al., 2015*) and downloaded from PATRIC (Table S1).

The taxonomy of the samples was verified using TYGS (*Meier-Kolthoff & Göker, 2019*). TYGS determines the closest related type strains using the MASH algorithm (*Ondov et al., 2016*) for entire genomes and BLASTn for 16S sequences. It calculates the pairwise distances of 16S and genome sequences using GBDP (*Meier-Kolthoff et al., 2013*), followed by inference of 16S and genome phylogenies based on the pairwise GBDP distances using FastME (*Lefort, Desper & Gascuel, 2015*), digital DNA-DNA hybridization (dDDH) using GGDC (*Meier-Kolthoff et al., 2013*), and deviation of G+C content. Genomes with >70% of dDDH and <1% of G+C content deviation are considered to be in the same species (*Meier-Kolthoff & Göker, 2019*).

## Positive selection analysis

A genome-scale positive selection analysis was performed using the PosiGene pipeline (*Sahm et al., 2017*) on the *Corynebacterium* genomes. Ten foreground genomes were tested (Table S2) representing 10 target clades or subclades. Six clades represent the species from the *C. diphtheria* complex (*C. belfantii*, *C. diphtheriae*, *C. pseudotuberculosis*, *C. rouxii*, *C. silvaticum* and *C. ulcerans*), two subclades represent *C. pseudotuberculosis* (biovars equi and ovis), and two subclades representing *C. ulcerans* lineages (lineage 1 and 2).

In the module "create_catalog", homologous genes were identified using BLASTp (*Camacho et al., 2009*) with the best-bidirectional hit criterion (*Altenhoff & Dessimoz, 2009*). In the module "alignments", orthologous genes are identified, gene trees are built, and a species tree is built from the gene trees. In this module, the anchor species is the genome that the orthologous gene sequences are aligned to, and the reference species is the genome from which the gene names are extracted. *C. diphtheriae* NCTC 11397[T] was selected as both the reference and anchor genome. In the first step, orthologous gene sequences were aligned to the anchor genome sequences using CLUSTALW (*Larkin et al., 2007*), with the parameters' minimum identity and minimum pairs identity set to 40%. The aligned sequences were filtered by GBLOCKS for gaps and unreliable alignment columns (*Jordan & Goldman, 2012*). In the next step, a phylogeny was built for each gene using the parsimony method and jackknifing implemented in DNAPARS from the PHYLIP package (*Felsenstein, 2005*). In the third step, a species tree was built based on the gene trees consensus, using CONSENSE from the PHYLIP package. The species tree is required to test for positive selection along specific lineages (*Yang & Nielsen, 2002*). The consensus tree was visualized using FigTree v1.4.4 and rooted using *C. kroppenstedtii* DSM 44385 as an outgroup. This strain was identified as an outgroup based on another tree generated by the same pipeline including *M. tuberculosis* H37RV to find the correct *Corynebacterium* species to use as an outgroup (Table S1) in that tree. The *M. tuberculosis* tree was not included in the downstream analysis.

In the module "positive_selection", a likelihood ratio test compares the non-synonymous to synonymous substitution rate $\omega = d_N/d_S$ in the foreground and the background. Here, $d_N$ is the number of non-synonymous substitutions per non-synonymous site and $d_S$ is the number of synonymous substitutions per synonymous site. The episodic positive selection model assumes $\omega > 1$ in the foreground and $\omega = 0$ or $\omega < 1$ in the background, while the null model assumes $\omega = 0$ or $\omega < 1$ for foreground and

background (*Yang & Dos Reis, 2011*). We considered positive selection if $p < 0.05$ for False Discovery Rate, as this correction is more suitable for genome wide experiments (*Storey & Tibshirani, 2003*).

Due to an assumption of no recombination by the branch-site models we used (*Yang, 2005*), recombination could cause false positive results (*Anisimova, Nielsen & Yang, 2003*). To avoid that artifact, the genes identified as positively selected by PosiGene were then tested for intragenic recombination using PhiPack (*Bruen, Philippe & Bryant, 2006*) that calculates Pairwise Homoplasy Index method (PHI) (*Bruen, Philippe & Bryant, 2006*), Neighbor Similarity Score (NSS) (*Jakobsen & Easteal, 1996*), and Maximum Chi-Square (*Smith, 1992*). In our analysis, we considered that recombination had occurred when $q < 0.05$ for PHI and at least NSS or Maximum Chi-Square (*Hongo et al., 2015*). Genes identified as recombinant were discarded for downstream analysis.

To minimize the false positive results caused by misalignments, frameshifts and ortholog prediction (*Schneider et al., 2009*; *Markova-Raina & Petrov, 2011*), we visually checked the alignments and checked the homology prediction by comparing the orthologs protein domains using PATRIC's annotation of Local and Global Families (*Davis et al., 2016*) and the Conserved Domain Database (CDD) (*Lu et al., 2020*).

## Gene annotation

Gene function was predicted using annotations from PATRIC and eggNOG-mapper (*Huerta-Cepas et al., 2017*), and InterProScan (*Jones et al., 2014*) was used to examine protein domains. Subcellular localization of the proteins was assessed using SufG+ v1.2.1 (*Barinov et al., 2009*). Protter (*Omasits et al., 2014*) was used to identify the position of the positively selected amino acids in relation to the cytoplasmic, transmembrane, and surface exposed portions of the proteins.

GIPSy v1.1.2 (*Soares et al., 2016*) was used to identify genomic islands (GIs) in *C. diphtheriae* NCTC 11397[T] using *C. glutamicum* ATCC 1302 (NC_006958.1) as the non-pathogenic reference. The islands were predicted by genes with C+G content and codon usage deviation, presence of transposases, presence of specific genes, non-conservation in comparison to reference genome, and flanking tRNA genes. The islands were classified according to the proportion of specific genes as pathogenicity islands, metabolic islands, resistance islands and symbiotic islands. Finally, prophages were predicted using PHASTER (*Arndt et al., 2016*).

## Prediction of drug and vaccine targets

Virulence genes and drug targets were predicted using the Pipeline Builder for Identification of Targets (PBIT) (*Shende et al., 2016*). PBIT predicted drug targets among the cytoplasmic proteins by a subtractive approach, identifying sequences of interest using BLASTp and specific databases in the following way. First, homologs to the human proteome, anti-targets and gut microbiota proteomes were filtering out to avoid cross-reactivity of drugs. Then the essential genes were identified using the Database of Essential Genes (*Zhang, 2004*) and virulence genes using the Virulence Factor Database (*Chen et al., 2005*). The druggability of the remaining candidates was predicted by

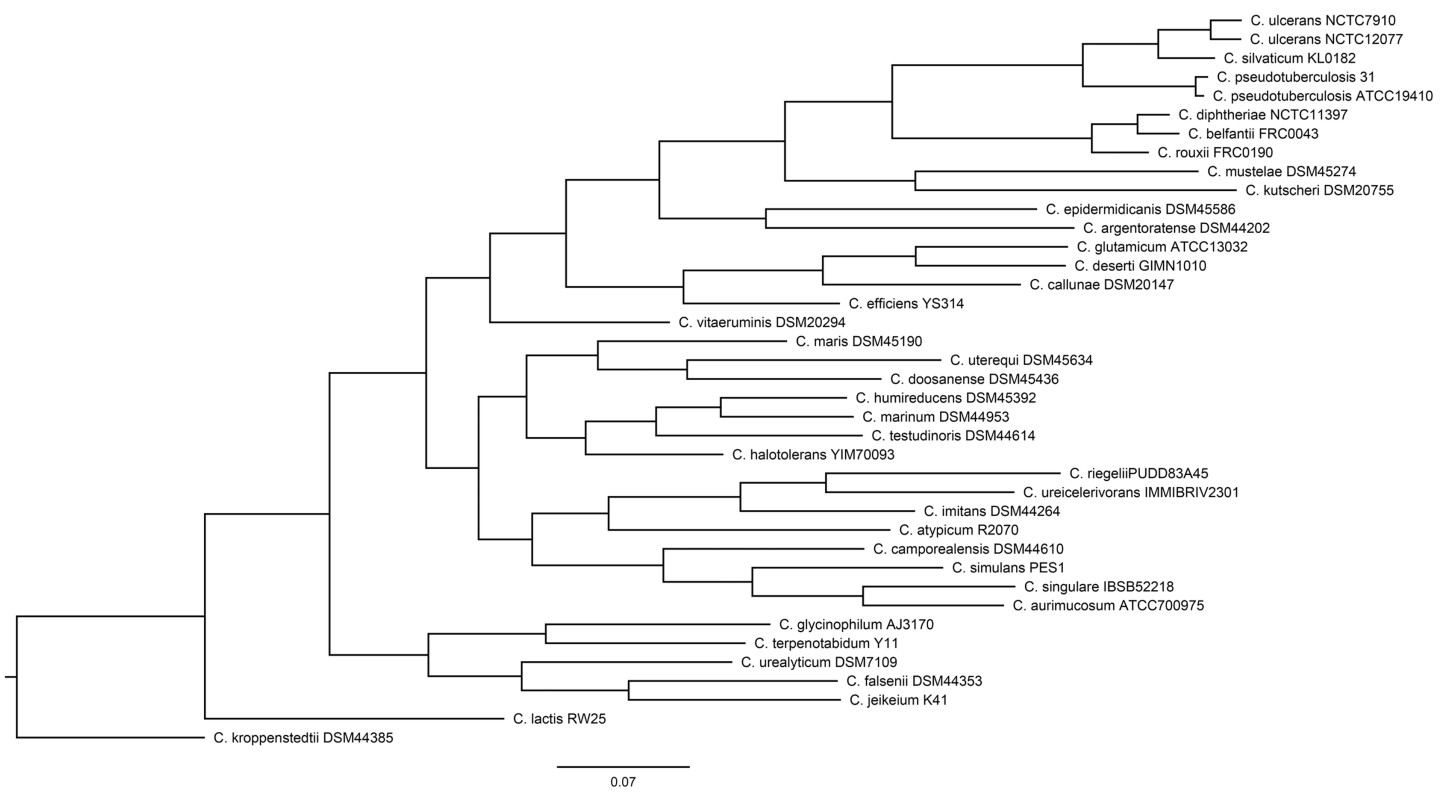

**Figure 1** The *Corynebacterium* species tree that was generated by the PosiGene pipeline, using CONSENSE from the PHYLIP package.

similarity to experimentally validated druggable targets from the Therapeutic Target Database (*Li et al., 2018*).

Vaccine targets were predicted from the transmembrane, putative surface exposed and secreted proteins predicted using SurfG+ and Vaxign (*He, Xiang & Mobley, 2010*) with prediction of human MHC Class I and II epitopes.

## RESULTS

Of the 40 genomes examined, the genome of *C. casei* LMG S-19264 was discarded due to its being identified as *Brevibacterium linens* based on the TYGS pipeline (Data S1). For this reason, only 39 *Corynebacterium* genomes were used for the downstream analysis. Figure 1 shows the phylogeny of the *Corynebacterium* genomes built by the PosiGene pipeline.

The PosiGene analysis showed zero to nine genes under positive selection ($p < 0.05$ for FDR) depending upon the 10 foregrounds that were used (Cb, Cd, Cp, Cpequi, Cpovis, Cr, Cs, Cul, Cul1, Cul2 (Tables S3–S12)), with 22 total genes shown to be under selection. Additional analysis of the 22 genes using PhiPack ($q < 0.05$ for PHI and at least NSS or Maximum Chi-Square tests) showed recombination in two out of four genes when Cd was the foreground, and one out of nine genes when Cp was used (Table S13). Manual curation of the 19 remaining proteins did not reveal any false positives caused by misalignments and ortholog prediction by comparison of protein domains (Table S14).

SurfG+ predicted 15 cytoplasmic, two membrane, one putative surface exposed and one secreted protein (Table S15). Thirty-five genomic islands and two prophages were predicted in *C. diphtheriae* NCTC 11397[T] (Tables S16 and S17).

From the 19 genes that were identified as positively selected across the species, two were identified in *C. diphtheriae*, 10 in *C. pseudotuberculosis*, one in *C. rouxii*, and six in *C. ulcerans* (Table 1 and Table S15). The COG categories of those genes were shown to be involved in defense, translation, energy production, and transport and metabolism of carbohydrates, amino acids, nucleotides, and coenzymes (Fig. 2). Based on our *in silico* prediction, 14 genes were found to be essential, six were virulence factors, and three were found to be in genomic islands (Table 1 and Table S15).

Thirteen of the genes were identified as potential drug targets by different analyses. Three genes were predicted based on our pipeline (Tables 1 and 2, and Table S15). The other 10 genes were not tagged as potential targets by the homology or druggability filters of the pipeline, but were included due to the possibility of targeting them with other methods (see Discussion section). For vaccine targets, four genes were predicted based on our pipeline (Tables 1 and 2, and Table S15).

## DISCUSSION

We identified 19 genes under positive selection, 13 potential drug targets and four potential vaccine targets. From the 13 potential drug targets, 10 were includes despite not passing the homology or druggability filters of the PBIT pipeline (Tables 1 and 2, and Table S15). The problem of having homology with the human proteome or human gut microbiota proteome can be solved by screening compounds that selectively inhibit the pathogen protein (*Arya et al., 2015*). The druggability prediction of the PBIT pipeline is based on sequence similarity to experimentally validated targets (*Shende et al., 2016*). So, the lack of predicted druggability by that pipeline can be solved by prediction of druggable pockets of a protein based on its own structure (*Volkamer et al., 2012*). Additionally, some of those proteins are known drug targets in other species.

### C. diphtheriae

In *C. diphtheriae* (foreground Cd), we identified two genes encoding proteins predicted to be essential and involved in translation, amino acid transport and metabolism (Table 1 and Table S15). The gene *ansB* is in GI10 and encodes secreted L-asparaginase type II which is a high-affinity enzyme that catalyzes the conversion of L-asparagine to L-aspartate and ammonia. The *E. coli*, *Dickeya dadantii* and human homologs to this gene are used for leukemia treatment, where the consequent low L-asparagine levels in plasma leads to apoptosis of the leukemia cells (*Lubkowski & Wlodawer, 2021*). This gene was suggested as a candidate vaccine target due to its classification as a secreted protein and predicted epitopes (Table 2). The second protein, SSU ribosomal protein S3p (*rpsC*), is a 30S ribosomal subunit that binds to the initiator Met-tRNA (*Burd & Dreyfuss, 1994*). Possible reasons for the selective pressure on these genes could be the effects on L-aspartate uptake (*ansB*) and translation efficiency (*rpsC*). *rpsC* was suggested as a drug target but

**Table 1 Characterization and possible application of 19 genes under positive selection in different species of the *Corynebacterium diphtheria* complex.**

| n | Product (Gene) | PS sites | PS positions | Local | COG | Essential | VF | Target | GenBank ID |
|---|---|---|---|---|---|---|---|---|---|
| | ***C. diphtheriae*** | | | | | | | | |
| 1 | L-asparaginase, type II (EC 3.5.1.1) (*ansB*) | 3 | 69, 182, 339 | S | EJ | Yes | No | Va[2] | ERS451417_00414 |
| 2 | SSU ribosomal protein S3p (S3e) (*rpsC*) | 1 | 89 | C | J | Yes | No | Dr[4] | ERS451417_00402 |
| | ***C. pseudotuberculosis*** | | | | | | | | |
| 3 | ABC transporter, permease protein (*mntC*) | 1 | 111 | M | P | Yes | Yes | Va[2] | ERS451417_00548 |
| 4 | Adenosine deaminase (*add*) | 1 | 25 | C | F | – | No | – | ERS451417_00570 |
| 5 | Adhesin SpaE (*spaE*) | 20 | 23, 33, 35, 108, 119, 122, 125, 223, 232, 238, 239, 243, 244, 247, 251, 252, 253, 255, 257, 259 | SE | – | No | Yes | Va[2] | ERS451417_00159 |
| 6 | Dihydropteroate synthase 2 (nonfunctional) (*folP*) | 2 | 135, 158 | C | H | Yes | No | Dr[1,3] | ERS451417_00887 |
| 7 | HNH endonuclease | 4 | 48, 111, 284, 352 | C | V | Yes | No | Dr[4] | ERS451417_00880 |
| 8 | Peptide chain release factor 1 (*prfA*) | 1 | 60 | C | J | Yes | No | Dr[3] | ERS451417_00951 |
| 9 | Putative oxidoreductase | 8 | 33, 42, 48, 52, 109, 209, 226, 285 | C | CH | No | Yes | – | ERS451417_02135 |
| 10 | Putative phosphoglycerate mutase (*pgmB*) | 2 | 69, 143 | C | G | Yes | No | Dr[1,3] | ERS451417_02267 |
| | ***C. pseudotuberculosis* equi** | | | | | | | | |
| 11 | Methionine aminopeptidase (EC 3.4.11.18) (*mapB*) | 2 | 96, 97 | C | E | Yes | No | Dr[3] | ERS451417_01521 |
| 12 | Tyrosyl-tRNA synthetase (EC 6.1.1.1) (*tyrS*) | 4 | 23, 58, 59, 403 | V | J | Yes | Yes | Dr[3] | ERS451417_01169 |
| | ***C. rouxii*** | | | | | | | | |
| 13 | Hypothetical protein | 4 | 5, 74, 95, 154 | M | S | – | No | Va[2] | ERS451417_00470 |
| | ***C. ulcerans*** | | | | | | | | |
| 14 | Serine hydroxymethyltransferase (EC 2.1.2.1) (*glyA*) | 1 | 385 | C | E | Yes | No | Dr[4] | ERS451417_00836 |
| | ***C. ulcerans* lineage 1** | | | | | | | | |
| 15 | Hypothetical protein | 1 | 32 | C | – | – | – | Dr[4] | ERS451417_00635 |
| 16 | Phosphoenolpyruvate-dihydroxyacetone phosphotransferase, dihydroxyacetone binding subunit DhaK (*dnaK*) | 4 | 248, 251, 255, 256 | C | G | Yes | Yes | Dr[4] | ERS451417_02360 |
| 17 | Similar to citrate lyase beta chain, 3 (*citE*) | 1 | 235 | C | G | Yes | Yes | Dr[1] | ERS451417_00750 |
| | ***C. ulcerans* lineage 2** | | | | | | | | |
| 18 | DNA polymerase III epsilon subunit (EC 2.7.7.7) (*dnaQ*) | 1 | 142 | C | L | Yes | No | Dr[4] | ERS451417_00985 |
| 19 | Precorrin-6A reductase (EC 1.3.1.54) (*cobK*) | 1 | 206 | C | H | Yes | No | Dr[4] | ERS451417_01234 |

**Notes:**

Columns: VF, virulence factor; COG, Clusters of Orthologous Groups; PS sites, positively selected sites; SE, surface exposed (sites).

Column Local: C, cytoplasm; M, membrane; SE, surface exposed; S, secreted.

Column COG: C, energy production and conversion; E, amino acid transport and metabolism; F, nucleotide transport and metabolism; G, carbohydrate transport and metabolism; H, coenzyme transport and metabolism; J, translation, ribosomal structure and biogenesis; S, function unknown; V, defense mechanisms.

Column Target: Dr, drug target; Va, vaccine target; [1]–predicted by PBIT pipeline, [2]–predicted by essentiality, local and Vaxign, [3]–described in literature for other species, [4]–suggested despite not attending one or more pipeline filters.

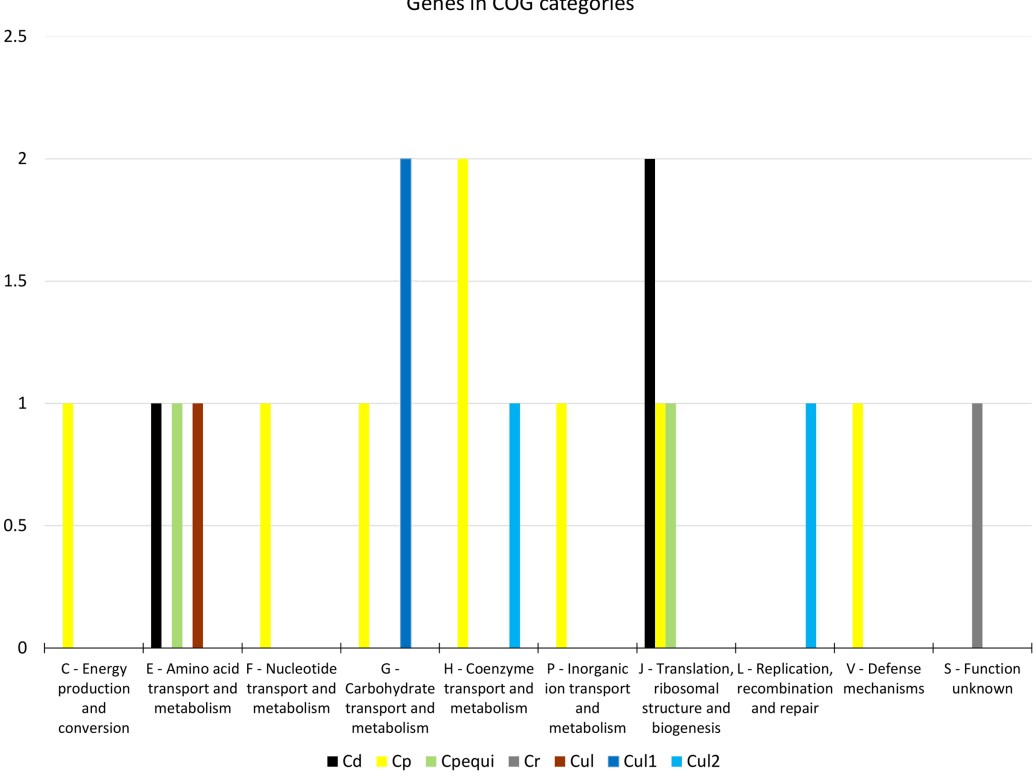

**Figure 2 Distribution of 19 genes under positive selection in COG categories.** Target groups: Cd, *C. diphtheriae*; Cp, *C. pseudotuberculosis*; Cpequi, *C. pseudotuberculosis* biovar equi; Cr, *C. rouxii*; Cul, *C. ulcerans*; Cul1, *C. ulcerans* lineage 1; Cul2, *C. ulcerans* lineage 2. COG categories: C-Q, metabolism; J-L, information storage and processing; M-V, cellular processes and signaling; S, poorly characterized.

has homology to a protein in the human gut microbiota proteome, requiring compounds that can selectively inhibit it.

### C. pseudotuberculosis

When *C. pseudotuberculosis* was the foreground (Cp), eight positively selected genes were identified. Five of the genes were tagged as essential, and three as virulence factors. They are involved in translation, coenzyme transport and metabolism, inorganic ion transport, defense from foreign DNA, nucleotide transport and metabolism and adhesion (Table 1 and Table S15). Among the essential genes, an ABC transporter permease protein (*mntC*) plays a role in the transport of $Mn^{2+}$ and $Zn^{2+}$ (*Claverys, 2001*) and was classified as a virulence factor. The Dihydropteroate synthase 2 (*folP2*) is nonfunctional according to PATRIC annotation but functional genes with this annotation are essential for the *de novo* synthesis of folate (*Bertacine Dias et al., 2018*). The HNH endonucleases degrade foreign DNA, and can also be involved in DNA repair, replication and recombination (*Wu, Lin & Yuan, 2020*). Peptide chain release factor 1 (*prfA*) recognizes the stop codons UAA and UAG, promoting the end of translation (*Scolnick et al., 1968*). The Putative phosphoglycerate mutase (*pgmB*) is capable of interconverting

**Table 2 Final drug and vaccine target candidates for *Corynebacterium* species based on a positive selection analysis by foreground, application, and priority.**

| Foreground | Application | Priority | Product (Gene) | PS sites (exposed sites) | Local | GenBank ID |
|---|---|---|---|---|---|---|
| Cd | Drug target | 1 (gut microbiota homolog) | SSU ribosomal protein S3p (S3e) (*rpsC*) | 89 | C | ERS451417_00402 |
| Cd | Vaccine | 1 | L-asparaginase (*ansB*) | 69, 182, 339 (69, 182, 339) | S | ERS451417_00414 |
| Cp | Drug target | 1 | Putative phosphoglycerate mutase (*pgmB*) | 69, 143 | C | ERS451417_02267 |
| Cp | Drug target | 2 (predicted as nonfunctional) | Dihydropteroate synthase 2 (nonfunctional) (*folP2*) | 135, 158 | C | ERS451417_00887 |
| Cp | Drug target | 3 (human and gut microbiota homolog, no predicted druggability) | Peptide chain release factor 1 (*prfA*) | 60 | C | ERS451417_00951 |
| Cp | Drug target | 4 (no predicted druggability) | HNH endonuclease | 48, 111, 284, 352 | C | ERS451417_00880 |
| Cp | Vaccine | 1 | Adhesin SpaE (*spaE*) | 23, 33, 35, 108, 119, 122, 125, 223, 232, 238, 239, 243, 244, 247, 251, 252, 253, 255, 257, 259 (23, 33, 35, 108, 119, 122, 125, 223) | SE | ERS451417_00159 |
| Cp | Vaccine | 2 (no exposed PS site) | ABC transporter, permease protein (*mntC*) | 111 | M | ERS451417_00548 |
| Cpequi | Drug target | 1 (virulence factor, more PS sites, gut microbiota homolog, target in literature) | Tyrosyl-tRNA synthetase (*tyrS*) | 23, 58, 59, 403 | C | ERS451417_01169 |
| Cpequi | Drug target | 2 (less PS sites, gut microbiota homolog, target in literature) | Methionine aminopeptidase (*mapB*) | 96, 97 | C | ERS451417_01521 |
| Cr | Vaccine | 1 | Hypothetical protein | 5, 74, 95, 154 (154) | M | ERS451417_00470 |
| Cul | Drug target | 1 (human and gut microbiota homolog) | Serine hydroxymethyltransferase (*glyA*) | 385 | C | ERS451417_00836 |
| Cul1 | Drug target | 1 (virulence factor) | Similar to citrate lyase beta chain (*citE*) | 235 | C | ERS451417_00750 |
| Cul1 | Drug target | 2 (virulence factor, gut microbiota homolog) | Phosphoenolpyruvate-dihydroxyacetone phosphotransferase, dihydroxyacetone binding subunit DhaK (*dhaK*) | 248, 251, 255, 256 | – | ERS451417_02360 |
| Cul1 | Drug target | 3 (hypothetical protein, no predicted druggability) | Hypothetical protein | 32 | C | ERS451417_00635 |
| Cul2 | Drug target | Equal. No predicted druggability | DNA polymerase III epsilon subunit (*dnaQ*) | 142 | C | ERS451417_00985 |
| Cul2 | Drug target | Equal. No predicted druggability | Precorrin-6A reductase (*cobK*) | 206 | C | ERS451417_01234 |

**Note:**
PS sites, positively selected sites.

2- and 3-phosphoglycerate in glycolysis (*Rigden, 2008*), although this particular gene is annotated as putative.

The other three identified genes (*add, spaE* and the putative oxidoreductase) are not characterized as essential, so may not be suitable drug targets. Adenosine deaminase (*add*) catalyzes the hydrolytic deamination of adenosine into inosine (*Chang et al., 1991*). *spaE*, from the operon *spaDEF*, encodes the minor pilin SpaE in *C. diphtheriae* (*Mandlik et al., 2008*) and is in GI5. It's ortholog in *C. pseudotuberculosis* also encodes the minor pilin and was first described as *spaB* from the operon *spaABC* (*Trost et al., 2010*). The putative oxidoreductase is a flavoenzyme with a "FAD-binding domain, ferredoxin reductase-type" (IPR017927), but its specific reaction is unknown.

Why would these particular genes be under positive selection? One could hypothesize that there would be more efficient manganese uptake (*mntC*), tissue adhesion on a new host range (*spaE*), improved efficiency for defense against foreign DNA (HNH endonuclease), translation (*prfA*), and metabolism of nucleotides (*add*) and carbohydrates (*pgmB*).

The vaccine targets (*mntC* and *spaE*) were indicated due to either their membrane location, their predicted epitopes, and that they might have surface exposed sites that are under positive selection. There were four drug targets (Table 2). *pgmB* was predicted as a target by PBIT and is this same gene is a drug target in helminth parasites (*Timson, 2016*). *folP2* was also predicted and is a well know target of sulfa and imidazole derivatives in human pathogens such as *Staphylococcus aureus*, *M. tuberculosis*, *Bacillus anthracis*, *Streptococcus pneumoniae*, *Burkholderia cenocepacia* and *Yersinia pestis* (*Bertacine Dias et al., 2018*). Although it was annotated as non-functional, the evidence of positive selection in this protein suggests an unknown adaptation due to specific amino acids that could be targeted. *prfA* has homology to human and human gut microbiota proteome and has no predicted druggability, but it is a known target of the drug Apidaecin in gram negative bacteria (*Matsumoto et al., 2017*). The HNH endonuclease had no predicted druggability.

### *C. pseudotuberculosis* biovar equi

When *C. pseudotuberculosis* biovar equi was used as the foreground (Cpequi), two genes were identified as being under positive selection, and were also characterized as essential. These two genes (*mapB* and *tyrS*) are both involved in translation (Table 1 and Table S15). Methionine aminopeptidase (*mapB*) cleaves the initiator methionine from newly synthesized polypeptides (*Helgren et al., 2016*; *Pillalamarri et al., 2021*). Tyrosyl-tRNA synthetase (*tyrS*) attaches the amino acid tyrosine to the appropriate tRNA (*Hughes et al., 2020*; *Othman et al., 2021*). These two genes could be under positive selection as it could affect translation efficiency. Both genes were predicted as homologs to human gut microbiota and have been identified as drug targets in other studies (Table 2). *tyrS* was predicted as a virulence factor and an ortholog from *Pseudomonas aeruginosa* was found to be targeted by four drug-like compounds (*Hughes et al., 2020*), and a *S. aureus* ortholog to this gene was targeted by new pyrazolone and dipyrazolotriazine derivatives (*Othman et al., 2021*). *mapB* from *M. tuberculosis* and

*S. pneumoniae* were shown to be selectively targeted despite homology to human protein (*Krátký et al., 2012*; *Arya et al., 2015*).

### C. rouxii

A single hypothetical protein (ERS451417_00470) was identified when *C. rouxii* was used as the foreground (Cr). It had no predicted domains, but it could be a vaccine target candidate (Table 2) due to its transmembrane location and the surface exposed sites under positive selection.

### C. ulcerans

A single essential gene (*glyA*) was identified when *C. ulcerans* was used as the foreground (Cul). *glyA* encodes a serine hydroxymethyltransferase enzyme (Table 1 and Table S15). This gene is known to participate in the one-carbon metabolism of serine/glycine interconversion and also in the folate/methionine cycle (*Batool et al., 2020*), which could explain its being under selective pressure. This gene was also identified as a potential drug target (Table 2), but it does have homology to human and human gut microbiota proteome. It has been shown to play a key role in lysostaphin resistance in *Staphylococcus aureus* (*Batool et al., 2020*).

### C. ulcerans lineage 1

Three genes were identified when the *C. ulcerans* lineage 1 genome was used as the foreground (Cul1). Two of them were essential, involved in carbohydrate transport and metabolism (Table 1 and Table S15). The Phosphoenolpyruvate-dihydroxyacetone phosphotransferase, dihydroxyacetone binding subunit DhaK (*dhaK*) gene is in GI34. This enzyme phosphorylates ketones and short chain aldoses using adenosine triphosphate (ATP) (*Peiro et al., 2019*). The second gene is annotated in PATRIC as "Similar to citrate lyase beta chain, 3" (*citE*) and is probably one of the catalytic subunits of citrate lyase, the enzyme that catalyzes the cleavage of citrate to acetate and oxaloacetate during citrate fermentation (*Schneider, Dimroth & Bott, 2000*). Both proteins were predicted as virulence factors by PBIT. The third gene encodes a hypothetical protein (ERS451417_00635) with no predicted domain or cellular localization.

The two genes with predicted function (*dhaK* and *citE*) appear to be related to metabolism inside the host. In *Listeria monocytogenes*, Phosphoenolpyruvate-dihydroxyacetone phosphotransferase (DhaK and other subunits) is required to utilize carbon sources for amino acid synthesis inside murine macrophages (*Eylert et al., 2008*). In *Enterococcus faecalis*, mutants of citrate fermentation genes (*citE* and others) were less pathogenic for the model *Galleria mellonella* (*Martino et al., 2018*). These same two genes are possible drug targets (Table 2). *citE* was predicted as a virulence factor and drug target candidate, while *dhaK* was predicted as a virulence factor and suggested despite the homology to a protein in the gut microbiota homology.

### C. ulcerans lineage 2

Two essential genes (Table 1 and Table S15) were identified when the *C. ulcerans* lineage 2 was used as the foreground (Cul2). The DNA polymerase III epsilon subunit (*dnaQ*) has a

domain with 3′–5′ exonuclease proofreading activity (*Raia, Delarue & Sauguet, 2019*). The other gene, *cobK*, encodes Precorrin-6A reductase which is involved in part I of the cobalamin cofactor (vitamin B12) biosynthesis pathway (*Kipkorir et al., 2021*). The mutations seen in these genes could provide the organism with a more efficient means of DNA replication (*dnaQ*) and biosynthesis of the essential cofactor cobalamin (*cobK*). Neither of these genes had any predictable druggability.

### Probable adaptations across groups

It is reported that most of the genes identified as being under positive selection are exposed on the surface and are involved in host colonization, and resistance to phage and antibiotics (*Petersen et al., 2007*; *Anisimova & Liberles, 2012*). Those under positive selection that are not surface exposed have been shown to be involved in metabolism (*Petersen et al., 2007*; *Rao, Sivakumar & Jayakumar, 2019*) or gene regulation (*Zhang et al., 2011*). Considering the function of the identified genes, most of the probable adaptations appear to be related to metabolism. A notable exception is in *C. pseudotuberculosis*, where the pilin SpaE could have enhanced adhesion to different host species tissues. Although the specific adaptations are not clear, an amino acid fixed by positive selection is an attractive target for a therapeutic molecule, as a non-synonymous mutation that could avoid interaction would decrease fitness. These genes could be used for reverse vaccinology and *in silico* drug targeting methods.

## CONCLUSION

In this analysis, we predicted 19 genes with non-synonymous mutations that are probably involved in adaptations found in the pathogens *C. diphtheriae*, *C. pseudotuberculosis*, *C. rouxii* and *C. ulcerans*. Based on our pipeline and literature data, 13 genes are candidate drug targets and four are potential vaccine targets, but their effectiveness would require experimental validation.

### Funding

This work was supported by CAPES (Coordenação de Aperfeiçoamento de Pessoal de Nível Superior, Brasil), CNPq (Conselho Nacional de Desenvolvimento Científico e Tecnológico) and FAPEMIG (Fundação de Amparo à Pesquisa de Minas Gerais). The funders had no role in study design, data collection and analysis, decision to publish, or preparation of the manuscript.

### Grant Disclosures

The following grant information was disclosed by the authors:
Coordenação de Aperfeiçoamento de Pessoal de Nível Superior, Brasil.
Conselho Nacional de Desenvolvimento Científico e Tecnológico.
Fundação de Amparo à Pesquisa de Minas Gerais.
## Competing Interests

Debmalya Barh and Vasco Azevedo are Academic Editors for PeerJ.

## Author Contributions

- Marcus Vinicius Canário Viana conceived and designed the experiments, performed the experiments, analyzed the data, prepared figures and/or tables, authored or reviewed drafts of the paper, and approved the final draft.
- Rodrigo Profeta conceived and designed the experiments, performed the experiments, analyzed the data, authored or reviewed drafts of the paper, and approved the final draft.
- Janaína Canário Cerqueira conceived and designed the experiments, performed the experiments, analyzed the data, authored or reviewed drafts of the paper, and approved the final draft.
- Alice Rebecca Wattam conceived and designed the experiments, authored or reviewed drafts of the paper, and approved the final draft.
- Debmalya Barh conceived and designed the experiments, authored or reviewed drafts of the paper, and approved the final draft.
- Artur Silva conceived and designed the experiments, authored or reviewed drafts of the paper, and approved the final draft.
- Vasco Azevedo conceived and designed the experiments, authored or reviewed drafts of the paper, and approved the final draft.

## Data Availability

The raw data are the gene nucleotide sequences listed in Table S1 and Tables 1 and 2.

## Supplemental Information

Supplemental information for this article can be found online at http://dx.doi.org/10.7717/peerj.12662#supplemental-information.

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
