# Peer review of "Evidence of episodic positive selection in Corynebacterium diphtheriae complex of species and its implementations in identification of drug and vaccine targets"

_PeerJ, doi:10.7717/peerj.12662_

## Round 0.1 · original submission · Major Revisions

Dear Dr. Viana and colleagues:

Thanks for submitting your manuscript to PeerJ. I have now received two independent reviews of your work, and as you will see, the reviewers raised some concerns about the research. Despite this, these reviewers are optimistic about your work and the potential impact it will have on research studying diphtheria control measures. Thus, I encourage you to revise your manuscript, accordingly, taking into account all of the concerns raised by both reviewers.

Please fully explain your use of “diptheria” in the Introduction and stay consistent with its meaning throughout. Also, justify your use of taxa in your analyses. Ensure that your findings are thoroughly placed within the existing body of literature.

There are many suggestions by the reviewers that should greatly improve your manuscript.

Thus, I encourage you to revise your manuscript, accordingly, taking into account all of the concerns raised by the three reviewers.

Good luck with your revision,

-joe

Reviewer 1 ·

Basic reporting

1. The authors should describe the rationale for and information provided by each of the pipelines in plain language for readers to better understand the input and outputs of each step.
2. The potential roles for several genes of interest are very briefly discussed, and the putative function(s) of the proteins in the context of other species is not made clear. While references are made for each protein, the original species, and experiments from which the putative functions are derived, is not explicit.
Some examples:
Line 185-186: “This enzyme is a potent antileukemic agent.”
Lines: 257-260: The utilization of carbon sources in macrophages and citrate fermentation.
3. In the summary results of the Abstract, the authors note that 62 different genes were detected with 45 shared across all species. Lines 176-177 appear to reference the presence of 45 genes, but this information is not expanded upon. Further discussion of the 45 genes appears to be lacking in the Results and Discussion section and should be provided.
Minor concerns:
1. Lines 50-51: The assertion of the rise in Diphtheria attributed to C. ulcerans does not appear to be fully supported by the references provided. While Hacker et al. note that cases of Diphtheria are caused mostly by C. ulcerans in Western Europe, the most recent literature review cited (Truelove et al. 2020) notes that C. ulcerans rarely causes Diphtheria.
2. Table 1: The predictions for virulence factors and essentiality of the proteins are derived from PBIT based upon the Materials and Methods. While this approach may be useful in the broad sense, it should be clearly indicated that these are predictions based upon PBIT and not experimentally established.
3. Lines 197-199: The reference cited by the authors for the ABC transporter, permease protein (MntC) notes the potential for Mn and Zn import through these permease proteins, not Mg and Zn.
4. Lines 199-201: The authors do not expand upon the significance of a potentially nonfunctional gene and how this fits within their study for drug or vaccine targets. The rationale for proposing it as a legitimate drug target is unclear (Lines 218-220).
5. Line 247: …(Batool et al., 2020), but. - I suspect this is a typographical error.

Experimental design

1. Based my understanding of the methodology described, it seems that only one C. diphtheriae strain was considered (NTCT 11397) and the diversity of C. diphtheriae isolates has not been considered. If proteins from this approach are to be explored as vaccine candidates, this limited approach may not capture the targets which the authors propose to identify.
2. The text refers to a strain NTCT 11397, which is not a strain that is within the sequenced database. I believe the authors meant NCTC 11397 – this is correct within the spreadsheets.

Validity of the findings

1. The Staphylococcus aureus MntC vaccine component is not the same as the MntC that was identified in their approach. The S. aureus MntC is the substrate binding lipoprotein component of the ABC transporter system, not the permease component. The mechanism for immune evasion likely stems from S. aureus MntC binding plasminogen (PMID: 25409527) – a function that is not established for the permease protein(s). The finding of MntC and all discussion related to the S. aureus MntC should be rewritten, as appropriate.
2. Lines 209-210: While SpaE may be considered a minor pilin in C. diphtheriae NCTC13129, it is unclear if this observation holds true for other diphtheria strains or for other Corynebacteria. Furthermore, the conservation of SpaE across all species is unclear. The authors may find the information provided in studies by Trost et al. (PMID: 22505676) and Broadway et al. (PMID: 23772071) helpful.

Reviewer 2 ·

Basic reporting

The paper of Vinicius Canário Viana et al describes a study in to the evolution of pathogenic bacterial strains belonging to the Corynebacterium diphtheriae group and identify genes that may have undergone positive selection, and they suggest that these could be useful treatment/vaccine targets

The authors use ‘diphtheria’ group throughout….this is not widely accepted in the field, I’m not opposed to it, but it should at least be qualified at first use perhaps at first use in the abstract…

the study is framed in an acceptable manner and based on a plausible hypothesis

Experimental design

I would be interested to know why Mycobacterium tuberculosis was used as an out group in the phylogenetic analysis, one could argue that it is phylogenetically distinct from the genus and perhaps a species of Corynebacterium that was not part of the ‘diphtheria’ group would be better – perhaps C. glutamicum, or C kroppenstedtii such as used as the outgroup in the analysis for positive selection.

Can the authors comment on the potential for skewing of the positive selection analysis by the use of small numbers of genomes for some taxa? Or does Posigene correct for taxon ID?

Validity of the findings

The data seem robust and there is potential novely and impact from the data, but the interpretation is sometimes not clear cut.
I felt the section of the discussion that focusses on the potential novel vaccine/therapeutic targets rather fizzles out without really identifying key candidates and rationalizing these…the authors should attempt to revise this with clearer conclusions. I was initially very excited by the title of the paper, but I’m not sure the conclusions are really born out by the way the final manuscript is written.
There are a few papers that talk about gene content in various biovars (sangal et al) and the work describing C belfanti that could be useful in framing some of the data...do any of the genes identified map to historical phenotypic differences used for biovar differentiation?

Additional comments

The whole manuscript should be revised for grammar and syntax by a proficient English language speaker to ensure the sentences reflect what the authors mean.

Line 82 – ‘fix’ should be ‘can help fix’ – positive selection doesn’t always fix the mutation – also the end of that sentence doesn’t make sense and should be reworded, I can’t help the authors as I’m not quite sure what they mean….

Line 167 ‘19 genes were identified’…as what…I am not sure what qualified these 19 genes?

182 – essential genes or proteins….as this is a bioinformatic study, they would be ‘genes encoding proteins predicted to…’

Line 221 – reference is J Timson 2016, should be corrected…

---

## Round 0.2 · accepted · Accept

Dear Dr. Viana and colleagues:

Thanks for revising your manuscript based on the concerns raised by the reviewers. I now believe that your manuscript is suitable for publication. Congratulations! I look forward to seeing this work in print, and I anticipate it being an important resource for groups studying diphtheria control measures. Thanks again for choosing PeerJ to publish such important work.

Best,

-joe

Reviewer 1 ·

Basic reporting

The authors have sufficiently addressed my comments. No further comments.

Experimental design

The authors have sufficiently addressed my comments. No further comments.

Validity of the findings

The authors have sufficiently addressed my comments. No further comments.